# Study of the Friction Behavior of Embedded Fibers in YG8 Surface Grooves

**DOI:** 10.3390/ma16145074

**Published:** 2023-07-18

**Authors:** Zhiping Huang, Haohan Zhang, Jing Ni, Lingqi Yang, Kai Feng

**Affiliations:** School of Mechanical Engineering, Hangzhou Dianzi University, Hangzhou 310018, China; huangzhiping2022@163.com (Z.H.); zhhh123@hdu.edu.cn (H.Z.); yanglqi@foxmail.com (L.Y.); fengkai319@foxmail.com (K.F.)

**Keywords:** nylon, YG8, textured surfaces, tribological characteristics

## Abstract

YG8 is a common cemented carbide material with excellent mechanical properties and mechanical properties, so it is widely used in the actual industry. However, due to the active chemical properties and strong affinity of tungsten alloy steel, it is easy to produce bonding and peeling in application, resulting in an unstable process and short service life. In order to control and reduce the surface wear of YG8 cemented carbide, groove-textured surface (GS) and flocking surface (FS) were prepared on smooth surface (SS). The friction characteristics of the samples were studied under different applied load conditions. The results show that the average friction coefficient of SS, GS and FS is inversely proportional to the load in dry/oil environment. Compared with SS, FS exhibits the lowest friction coefficient, which is reduced by 30.78% (dry friction) and 13.13% (oil lubrication). FS effectively improves the tooth jump phenomenon of the sample and the amplitude of the friction coefficient, friction force and load, and has the best anti-friction characteristics. At the same time, the FS with the fastest contact angle drop at any time also showed excellent wetting ability, and the wear rate decreased by an order of magnitude. The implantation of fibers in the groove inhibits the spalling and furrow of wear track, which is attributed to the effect of fibers on damage repair. In the friction process, FS increases the content of the O element and induces the formation of oxides. The friction mechanism is mainly chemical wear. The excellent tribological properties of FS have a good guiding significance and theoretical support for improving the tribological properties of high hardness material surfaces.

## 1. Introduction

As an important component of an aerospace engine core equipment cutting tool, the excellent mechanical properties and performance of cemented carbide (YG8) is an important factor in ensuring the cutting performance of the tool [1,2]. However, due to the complex working environment, turbine discs are subjected to high temperature, high pressure, high centrifugal force and other combined effects of alternating loads for a long time; the cutting tool should have the reliability necessary for long-term service to ensure stable machining accuracy [3,4]. Furthermore, there is a requirement that tool materials in the complex working environment should have excellent wear resistance and wetting characteristics [5,6,7]. Therefore, it is significant to study the friction characteristics of tungsten steel to enhance the tool performance.

To effectively enhance the wear resistance and anti-friction properties of metals, the preparation of micro-textures on metal surfaces with special microscopic effects (microfluidic dynamic pressure effect [8,9], micro-trap of abrasive chips [10] and retention of lubricating fluid [11], etc.) has garnered wide attention in the field of tribology [12,13,14,15]. Yuan et al. [16] evaluated seven different angles of microgrooves on metal surfaces to study the effect of different orientations of microgrooves on friction properties. The experimental results showed that parallel and perpendicular grooves have a greater effect on the friction reduction of the frictional surface. In addition to grooved textures, other functional micro-textures also play an important role in friction reduction. To enhance the wear resistance of rolling bearings, Wu et al. [17] studied three different liquid self-driven micro-textures on the surface of a bearing. It was found that the gradient micro-texture has the best effect in terms of reducing friction and suppressing vibration. Wos et al. [18] prepared chevron-type textures on graphite surfaces by abrasive spray processing and then carried out friction experiments. The results showed that the chevron-type textures are very effective in reducing friction. In addition, Oh et al. [19] prepared V-grooves on the graphite surface which could effectively reduce the friction coefficient and prolong the service life of micro ball bearings. However, using micro-texture also leads to the material surface being prone to damage phenomena such as microcracking and spalling near the micro-texture.

Filling the friction interface with lubricant is also an effective way to improve the wear resistance of metal surfaces [20,21]. Also, the introduction of lubricants can promote the formation of oil films on metal surfaces [22,23], which enhances the tribological properties of metal surfaces. For example, Xiong et al. [24] used Eu doped with CaWO4 nanoparticles as an anti-friction material. It was found that Eu-doped CaWO4 nanofluid is a material with excellent load-bearing capacity, abrasion resistance and anti-frictional properties. He et al. [25] attached Al_2_O_3_ and MoS_2_ mixed nano-lubricants to metal plates to study the frictional behavior and synergistic lubrication mechanism of the two nanoparticles. The non-equilibrium molecular dynamics model revealed that Al_2_O_3_ promotes the rolling effect as well as enhances the interlayer sliding effect of MoS_2_ between metals under a water base, resulting in the lowest and most stable friction coefficient and wear rate. In addition, ZrO_2_ and TiO_2_ are two excellent nanoparticles, and Huang et al. [26] used ultrasound to mix the two materials with graphene solutions to form hybrid nanosuspensions. Using nanosuspensions reduces the abrasive wear and furrow phenomenon relative to smooth surfaces.

In recent years, there has been a lack of studies related to combining micro-texture with materials that promote lubrication [27], vibration suppression [28], noise reduction [29] and other related effects in order to achieve synergistic effects. Xue et al. [30] prepared a groove on a CSS-42L surface and filled it with Sn-Ag-Cu-Ti_3_C_2_ material to enhance the anti-friction properties and noise reduction performance of bearing steel. The results showed that the synergistic effect of the grooves and Sn-Ag-Cu-Ti_3_C_2_ enhances the repair ability of surface damage and improves the anti-friction properties. In addition, Xing et al. [31] combined micro-texture with WS_2_ solid lubricant to achieve the improved frictional wear performance of ceramic tools. Rapoport et al. [32] added MoS_2_ to the surface of laser micro-texture to form a composite lubricating surface, and the micro-texture continuously supplied solid lubricant to the friction surface, which effectively reduced the friction coefficient and improved the service life.

At present, there are few reports on the application of fiber materials to functional surfaces. Fiber materials have excellent elastic recovery and liquid adsorption ability. And fiber is a promising lubricant material in industrial applications. In this paper, groove-texture surface (GS) and flocking surface (FS) inside the fiber-filled grooves were prepared on the smooth surface (SS) of YG8. Theoretical studies and mechanistic analyses are conducted for the anti-friction characteristics, wetting characteristics and wear resistance of FS. In engineering applications, this research provides the theoretical and experimental support for YG8 materials to reduce frictional wear and improve wetting lubrication.

## 2. Experiments and Methods

### 2.1. Materials Reparation

PA fiber has good comprehensive properties, including mechanical properties, heat resistance, wear resistance, self-lubrication and low friction coefficient [33]. More importantly, PA is a conductive material that is conducive to implantation in metal surfaces. Therefore, PA (purchased from Hangzhou Yunmao Plastic Co., Ltd., Hangzhou, China) is selected as a material to improve the wear resistance of metal surfaces; material properties of PA are shown in Table 1. In addition, YG8 has a series of excellent characteristics such as high hardness and corrosion resistance, which are widely used in high-precision machining. The wear of YG8 will cause a huge impact on the workpiece to be processed. Therefore, YG8 (purchased from Wuhu Cemented Carbide Cutting Tools Co., Ltd., Wuhan, China) was selected as the sample material, and the material properties are shown in Table 2.

To meet the requirements of the Retc friction tester, the surface of each sample was carefully polished before the test with a roughness of 0.05 mm. Subsequently, a laser marking machine was used to prepare the grooved texture on the YG8 surface (see Figure 1a), and the parameters of the laser processing are shown in Table 3. The grooved texture of sample surface (see Figure 1b) was designed to ensure the ability to trap abrasive chips during testing [34], and the geometry of the test sample is shown in Table 4. The surface of each sample was polished with 2000# and 4000# sandpaper for 20 min because of the melt generated on the sample surface after laser etching. Finally, the samples were cleaned in an ultrasonic cleaner for 30 min.

PA fibers (polyamide) were implanted in the grooves using electrostatic flocking technology. Firstly, a layer of adhesive was applied to the grooves at room temperature (23 ± 2 °C), and the thickness of the adhesive was around 0.002–0.008 mm. Secondly, the samples were placed in the negative electrostatic flocking box, and the nylon material was placed in the positive electrode. The distance between the nylon material and the samples was 400 mm. The flocking time was 10 min and the specific parameters of electrostatic flocking are shown in Table 5. The prepared samples were placed in the chamber for 24 h and subsequently cleaned using an ultrasonic cleaner for 20 min to ensure that the fibers floating on the flocking surface were removed. The specific preparation procedure is shown in Figure 1c,d.

### 2.2. Experimental Procedure

The friction behavior of SS, GS and FS was investigated by a friction and wear tester (MFT-5000 Rtec, San Jose, CA, USA) under dry/oil conditions, as shown in Figure 2. The friction balls used were GCR15 steel balls with a diameter of 6.35 mm. Considering the friction characteristics of the samples and the actual working conditions, the experimental method was reciprocating friction with a displacement amplitude of 4.15 mm. The scheme of the detailed experiment is developed in Table 6. Given the randomness of the experiment, three repetitions of the experiment were implemented for each sample to ensure the reliability and accuracy of the experimental data. The specific working conditions for the friction and wear experiments are shown in Table 7.

The test system contains a high friction sensitivity mechanical sensor (MFT-5000 Rtec, USA) for collecting friction force, load and friction coefficient. The data are captured at a frequency of 100 values per minute. The data collected every second were taken as an average value of the friction coefficient for ease of analysis. Moreover, the maximum and minimum values for each experiment were included by setting the error bars.

In order to find out the influence of oil lubrication on the friction performance of FS, the wettability test of different samples was carried out by contact angle measuring instrument (JC2000D1, Shanghai Zhongchen Digital Technology Equipment Co., Ltd., Shanghai, China). The acquisition frequency of the contact intersection measuring device was set to 1 s to capture 9 images. The lubricant of FS was completely wetted after 8 s, so the images of the first 8 s were collected and analyzed.

After the test, the wear morphology of the experimental samples was collected using a high-speed camera (Type: KEYENCE VW-9000, purchased from KEYENCE Co., Ltd., Shanghai, China). Scanning electron microscopy (SEM, TSM-IT300/EDS-X-MaxN20, Japan) was used to observe the microstructure morphology and element distribution on the YG8 surface and for the analyzation of the wear mechanism. The roughness of the sample surface was measured by a roughness meter (Type: Mitutoyo SJ410, purchased from Suzhou Zesheng Precision Machinery Instrument Co., Ltd., Suzhou, China).

## 3. Results and Discussions

### 3.1. Tribological Properties of Samples

As shown in Figure 3, the average friction coefficients of SS, GS and FS all decreased with increasing applied load under different working conditions. In the load range of 60–120 N, oil lubrication made the friction coefficient of the samples fluctuate less than that of dry friction, which effectively improved the wear resistance of the sample. Compared with dry friction, the lubrication effect of oil enhanced the anti-friction characteristics of the metal surface. The friction coefficients of the samples (120 N) were reduced by 73.64% (SS), 76.10% (GS), and 66.91% (FS), respectively. The micro-hydrodynamic effect caused by oil effectively improved the wear resistance of the samples. In addition, the presence of the groove-textured surface (GS) curbed the wear resistance of the sample surface in all operating conditions, while FS showed better frictional characteristics. As shown in Figure 3b, compared to SS, the friction coefficient of GS (frequency of 1120 N) increased by 23.79% (dry friction) and 15.93% (oil lubrication), respectively, while the friction coefficient of FS (frequency of 1120 N) decreased by 30.78% (dry friction) and 13.13% (oil lubrication), respectively. FS exhibited the smallest friction coefficient in dry and oil-lubricated scenarios, with the best anti-friction characteristics at a load of 120 N.

Figure 4 shows the friction coefficients of the samples for different working conditions from 0 to 1200 s. Compared with dry friction, a smoother friction profile was obtained for the oil lubrication, with a decreasing trend in the friction profile. However, as shown in Figure 4c,d, the friction coefficient of GS produced a rapid peak followed by a slow decrease at around 50 s and 200 s, and the friction coefficient was always higher than that of SS. This is due to the fact that the edge of GS was constantly subjected to the impact of the counterpart ball during the friction process, resulting in serious wear on the surface [30]. As shown in Figure 4e, a similar condition occurred in FS under dry friction, but the friction coefficient of FS was always smaller than that of SS because the fibers filling the grooves enhanced the anti-friction effect on the metal surface. As shown in Figure 4f, the friction coefficient of FS was smoother and tended to decrease under oil lubrication due to the superior adsorption and storage capacity of the FS surface to the lubricant. FS showed the smallest fluctuation of friction coefficient amplitude under oil lubrication.

The reason for the increase in the friction coefficient of GS was that, on the one hand, the material was brittle due to the high hardness of YG8. Therefore, there was a constant impact with the grooved edge during the reciprocal friction of the steel balls, which made the grooved edge prone to fracture, changing the wear resistance of the metal surface [16]. On the other hand, it was because the size of the groove was of a deep submicron level, which breaks the continuity of the surface, and the friction process caused significant vibration and noise, which could not improve the wear resistance of the metal surface [31]. However, when fibers were implanted in the grooves, they were not only able to contain the reverse effect of the grooves and improve the bearing capacity of oil film, but also further enhance the anti-friction properties of the sample surface due to the excellent flexibility, rebound characteristics and wear resistance of the fibers.

### 3.2. Friction Behavior of Samples in Steady Stage

The differences in surface morphology lead to different fluctuations in friction coefficient, load and frictional force. Figure 5 presents the friction behavior of different samples during the stabilization phase (1100–1101 s). Compared to dry friction, lubricant improved the magnitude and variation of the curves, with the most obvious improvement in FS. As shown in Figure 5a–d, the grooves disrupted the continuity of the macroscopic surface during the friction process, making the ball move on the sample surface with different degrees of tooth jump. Because FS also had the same groove structure, FS had a similar situation. It is clear to see that the number of jumps of GS and FS in the friction behavior curves was the same as the number of grooves worn on the sample surface. Meanwhile, compared to SS, GS had the largest amount of fluctuation and FS had the smoothest. As shown in Figure 5c,e, under dry friction, the grooves caused severe fluctuations in the applied load, resulting in the largest value of the friction coefficient, and the smallest load fluctuation for FS. As shown in Figure 5d,f, under oil lubrication, the load curve fluctuations were all significantly reduced, where the load fluctuation of SS was greater than that of GS, which was due to the grooves causing a micro-hydrodynamic pressure effect under the action of oil, thus reducing the fluctuation of the load [8,9].

Reciprocal friction frequency and applied load seriously affected the frictional behavior of sample surfaces. Under dry/oil friction, the impact at the grooved edge and the tooth jump phenomenon aggravated the wear of the metal surface and the contact surface of the steel ball, which led to an increase in the friction coefficient [22]. However, the impact and tooth jump phenomena were not only alleviated by the promotion of fibers but also reduced the friction coefficient. In addition, the oil on the flocking surface induced a micro-hydrodynamic pressure effect under repeated compression and impact, which promoted the generation of an oil film and of bearing capacity on the metal surface. Under the compound lubrication system, the oil film generated by this effect could further enhance the anti-friction effect of the surface.

### 3.3. Wettability Behavior of Lubricants

As shown in Figure 6, the wetting ability of SS, GS and FS on the lubricant was analyzed, and the comparative study of the lubrication ability of different surfaces helps to provide theoretical support for engineering applications. The smaller the contact angle of surface droplets, the better the wetting ability. At room temperature (25 °C), when the droplet viscosity was constant, the contact angle between the oil and SS surface was the largest and the contact angle between the oil and FS was the smallest. Within 1–2 s of droplet contact, the contact angles of both GS and FS produced a large decrease, which was due to the pressure difference between the inside and outside of the groove [35]. As time increased, GS spread at a relatively stable rate. However, compared to SS and GS, FS still spread in all directions with a larger slope, and the droplets had spread almost completely on the surface at about 8 s. This is because, when the droplets enter the interior of the groove, the droplets first spread rapidly due to the Laplace pressure difference, and then the droplets spread further, owing to the capillary force between the fibers and the adsorption ability of the droplets by the fibers [36]. Therefore, FS exhibited the best wetting ability.

Compared with SS, GS had a better wetting ability, which just proves that the load fluctuation of the above GS was less than that of SS under oil lubrication. In addition, the implanted fibers in the groove effectively promoted the adsorption of lubricating oil on the surface, which was also an important reason for the fluctuation of the minimum friction behavior curve of FS. The outstanding wetting ability of FS could effectively enhance the adsorption and formation of oil film on the friction surface, indicating that the synergistic effect of the solid–liquid lubrication system shows a better promising application in the composite lubrication system.

### 3.4. Worn Surface Analysis

The wear rate of a sample was an important parameter in the tribology analysis that visually reflects the anti-friction characteristics and wear resistance of different friction surfaces (SS, GS and FS). The wear rate (δ) was calculated using the following equation:(1)$$\delta =\frac{V}{W}$$
where V denotes the wear volume and W denotes the friction work.

As shown in Figure 7, the wear rates of different friction surfaces (SS, GS and FS) at different frequencies were investigated. It is clear that the wear rates under oil lubrication (SS-O, GS-O, FS-O) were lower than those under dry friction (SS-D, GS-D, FS-D). Compared to dry friction (120 N), the wear rates of SS, GS and FS were, respectively, decreased by 39.68% (SS), 18.18% (GS) and 14.25% (FS), which is attributed to the excellent lubrication performance of the lubricant. Moreover, the wear rate increased with increasing applied load. In addition, the wear rates of GS and FS increased when compared with SS at a certain frequency and load. Under dry/oil lubrication, compared to SS, the wear rates increased by 45.52% (dry friction) and 28.38% (oil lubrication) for GS (120 N), respectively. The wear rates of FS increased by 41.69% (dry friction) and 17.09% (oil lubrication), respectively, in comparison to SS because, during the friction process, the counterpart ball and groove before the impact caused increased wear on the contact surface of the counterpart ball, which induced an increase in the contact area. However, as shown in Figure 7, compared to GS (120 N), the wear rates of FS were reduced by 13.61% (dry friction) and 6.5% (oil lubrication), respectively. This indicates that FS was effective in suppressing the wear rate of the surface.

During sliding, the grooves disrupted the continuity of friction. When the counterpart ball came into contact with the edge angle of the groove, it caused damage to the counterpart ball, resulting in an increase in the contact area between the counterpart ball and the friction surface, which led to an increase in the wear rate. However, the rebound characteristics of the fiber and the good material properties reduced the wear rate of the surface. Despite the wear rate of FS increasing, it maintained a significant tribological performance due to its excellent lubrication properties.

### 3.5. Surface Profile Characteristics

The surface roughness of the sample was analyzed. As shown in Figure 8, there was a uniform reduction in the roughness of the sample surfaces with increasing applied load for both the dry and oil-lubricated scenarios. Also, the roughness of the surfaces obtained under oil lubrication was lower than that of dry friction. Compared to dry, the surface roughness obtained by oil lubrication decreased by 37.50% (SS), 45.00% (GS) and 56.25% (FS) at 120 N, respectively, indicating that FS, with excellent lipophilicity, could significantly improve the surface roughness of the sample. As shown in Figure 8b, under oil lubrication, the surface roughness of GS decreased by 19.23% (60 N), 15% (80 N), 27.78% (100 N) and 26.67% (120 N), respectively, compared to SS, which is attributed to the role of GS in storing debris. In addition, the surface roughness of FS decreased by 42.31% (60 N), 40.00% (80 N), 55.56% (100 N) and 53.33% (120 N) compared to SS, which is consistent with the results discussed above. Therefore, compared to the other samples, FS had the smallest roughness value and showed the best anti-friction characteristics under dry/oil friction.

To understand the wear mechanism of FS, the wear morphology of the sample under 120 N load was analyzed. As shown in Figure 9, the morphology of the sample surface was presented under dry friction and oil lubrication. Compared with dry friction, oil lubrication significantly improved the wear morphology of the surface. As shown in Figure 9a–c, a large number of wear debris, furrows and a small number of pits appeared on the surface of SS under dry friction. GS with grooves could effectively store wear debris, but there were still many grooves. FS had only shallow grooves and oxidative exfoliation on the surface under the combined action of fibers and grooves. At the same time, under oil lubrication, the surface morphology of the sample also showed the same trend, among which the SS surface was the worst, and FS showed the best surface morphology, which is consistent with the results of the above discussion (as shown in Figure 9d–f). To reveal the friction mechanism of FS, the surface element composition of SS and FS after friction was analyzed, as shown in Figure 9g,h. Compared with SS, the atomic percentage of O element on the surface of FS increased significantly by 44.60% after friction, and the atomic percentage of C and W elements decreased to varying degrees. This is because the fibers of FS were mainly composed of N, H and O elements. During the friction process, the migration and diffusion of fiber elements promoted the distribution of O elements. More importantly, the increase in the O element contributed to the formation of oxides. The oxides could effectively improve the wear resistance and anti-friction characteristics of FS surface.

### 3.6. Antifriction Mechanisms of FS

Wetting and lubrication are one of the most effective ways to reduce wear and improve wear resistance. In this paper, we used laser processing technology to prepare a microstructure on the surface of the sample, and then used electrostatic flocking technology to implant fibers into the microstructure, so as to prepare a composite lubrication structure to improve the surface wear resistance. The friction mechanism of FS was clarified in terms of two aspects: microstructure and electrostatic flocking.

As shown in Figure 10, during the reciprocating friction process, the wear debris at the SS interface could not be removed in time, and the wear debris had a serious impact on the interface friction. Therefore, the untreated SS produced the most serious damage, and the friction mechanism was mainly abrasive wear. In Section 3.2, the wetting characteristics of GS were better than those of SS, which indicated that GS had a positive effect on improving the anti-friction characteristics. On the one hand, the groove structure could enhance the retention of the lubricant and improve the hydrodynamic effect of the microfluidic. On the other hand, the groove structure could store wear debris and reduce the wear caused by wear debris. The wear mechanism was mainly adhesive wear.

As shown in Figure 10c,d, the wear resistance of the sample surface was further improved after the fiber was implanted in the groove. FS could not only retain the advantages of the groove structure, but also make full use of the positive effects of the fiber. Because FS had good hydrophilicity, it could significantly improve the wetting characteristics of the surface, thereby reducing the impact of friction heat. In addition, the fiber had excellent rebound recovery ability. In the friction process, when the ball was in contact with the fiber, the fiber could remove the wear debris on the surface of the corresponding ball and slow down the negative effect of the wear debris on the interface. More importantly, the presence of fibers promoted the formation of surface oxides, and the wear mechanism was mainly chemical wear. Therefore, FS exhibited the best wear resistance and anti-friction characteristics.

## 4. Conclusions

In this paper, the effect of composite structures (grooves and fibers) was prepared on the material surface. The wear characteristics of YG8 were investigated, and the wetting characteristics of the composite lubrication structure were analyzed. Based on the above results, the summary is as follows:(1)The friction coefficient was positively correlated with the load, which was because the load changed the real contact area between the friction pairs. Among them, FS had the lowest friction coefficient in dry/oil environment, showing excellent wear resistance and anti-friction characteristics.(2)Among SS, GS and FS, GS disrupted the surface continuity with severe tooth jump and FS exhibited the lowest friction coefficient and friction load. The synergistic effect of the lubricant and composite structure significantly decreased the fluctuation of friction force and applied load, where FS exhibited the best anti-friction characteristics at 1 Hz and 120 N.(3)The adsorption ability of FS to droplets improved the retention and storage of lubricant and promoted the formation of oil film, thus enhancing the friction reduction performance of FS.(4)The wear mechanism of SS was abrasive wear, the wear mechanism of GS was adhesive wear and the friction mechanism of FS was chemical wear. FS improved the friction performance of the lubricant under chemical wear, repaired surface damage and improved wear resistance.(5)During the sliding process, the fiber material diffused and migrated at the friction interface, which accelerated the formation of an oxide layer. The FS surface effectively improved the surface morphology of the sample and exhibited excellent tribological performance.

## Figures and Tables

**Figure 1 materials-16-05074-f001:**
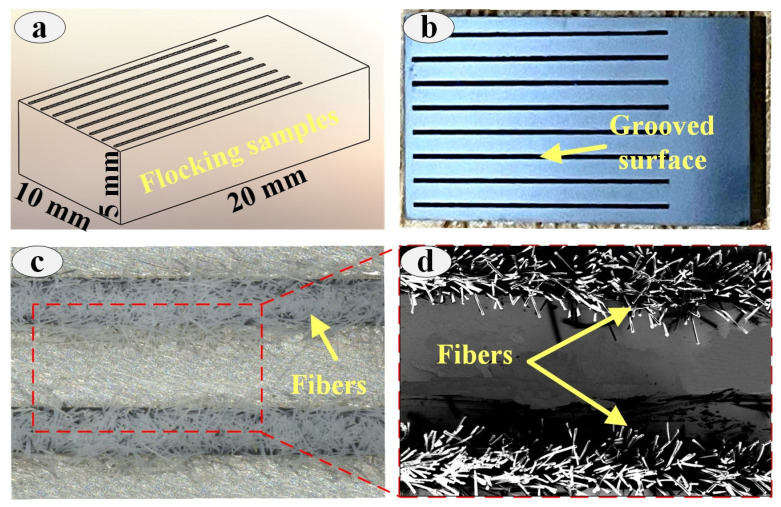
Sample size(**a**), grooved surface image (**b**) and flocking surface image (**c,d**).

**Figure 2 materials-16-05074-f002:**
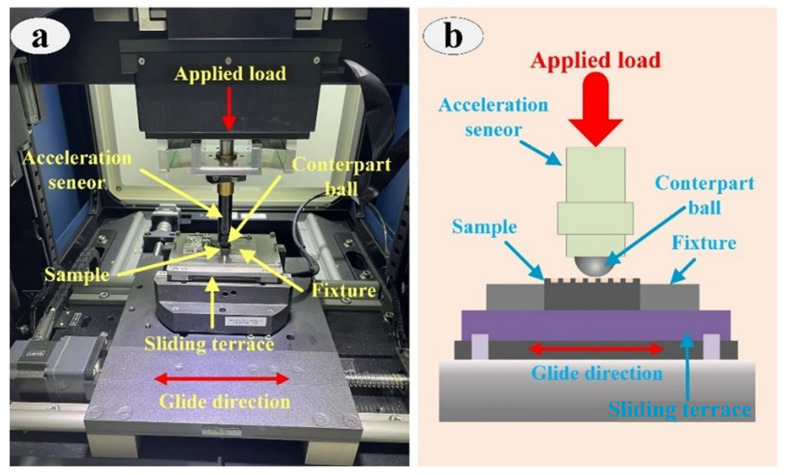
Photograph (**a**) and schematic (**b**) of the test system.

**Figure 3 materials-16-05074-f003:**
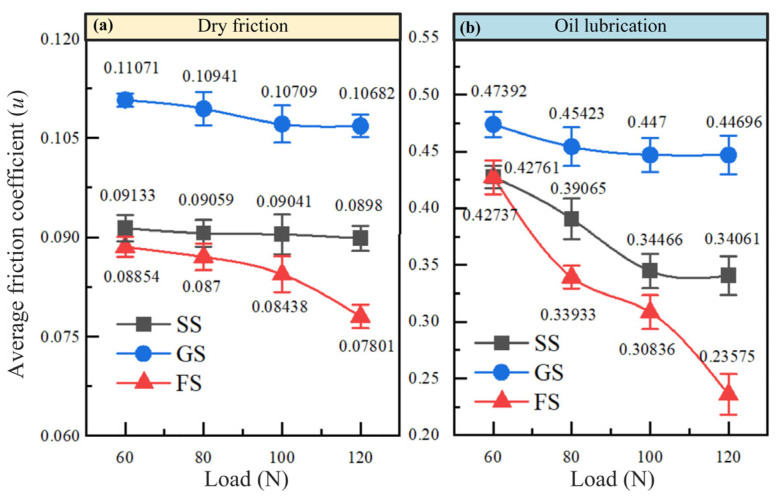
Comparison of average friction coefficients of samples under dry friction (**a**) and oil lubrication (**b**).

**Figure 4 materials-16-05074-f004:**
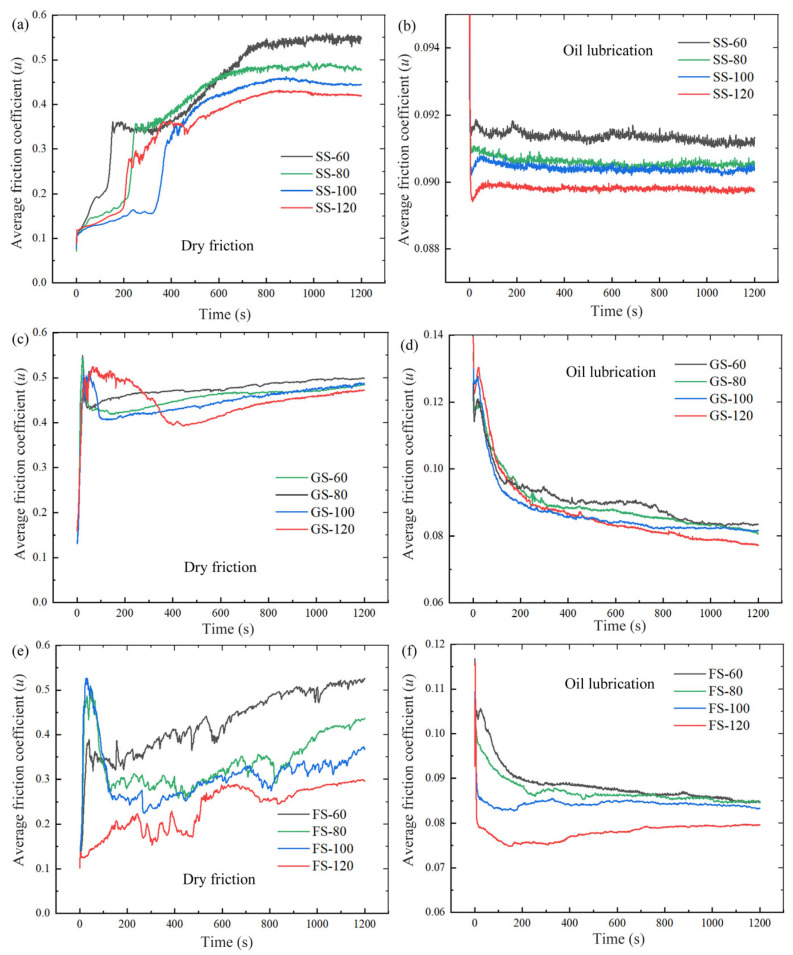
Friction coefficient of SS (**a**,**b**), GS (**c**,**d**), FS (**e**,**f**) under dry/oil friction.

**Figure 5 materials-16-05074-f005:**
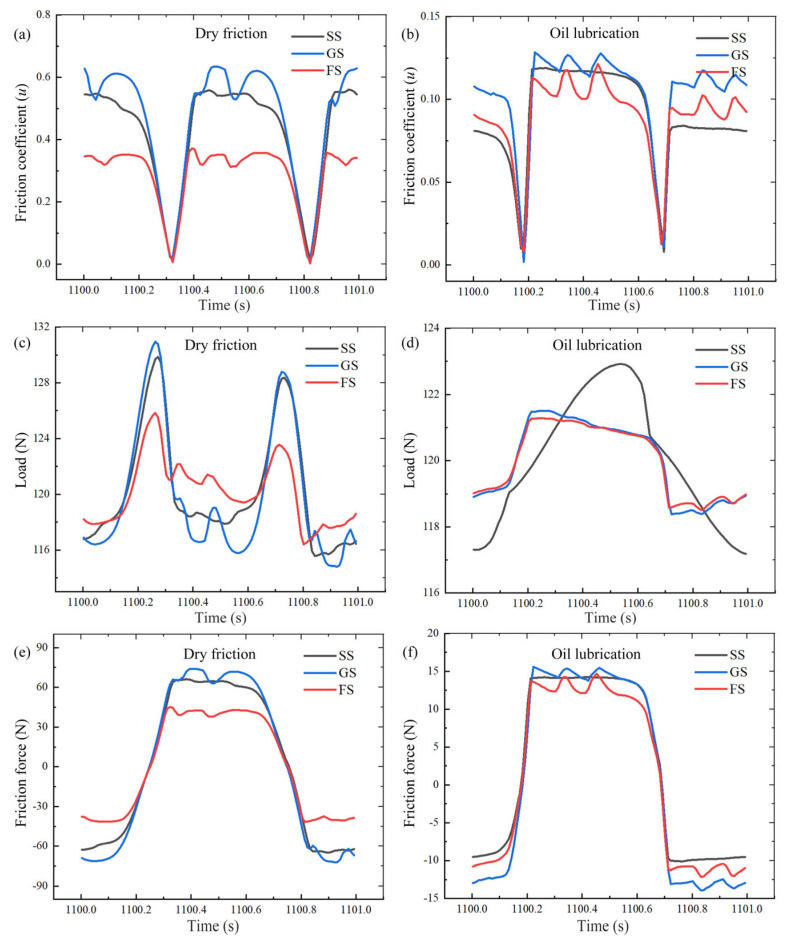
Changes in friction coefficients (**a**,**b**), loads (**c**,**d**) and friction forces (**e**,**f**) of samples with 120 N load at the stable stage.

**Figure 6 materials-16-05074-f006:**
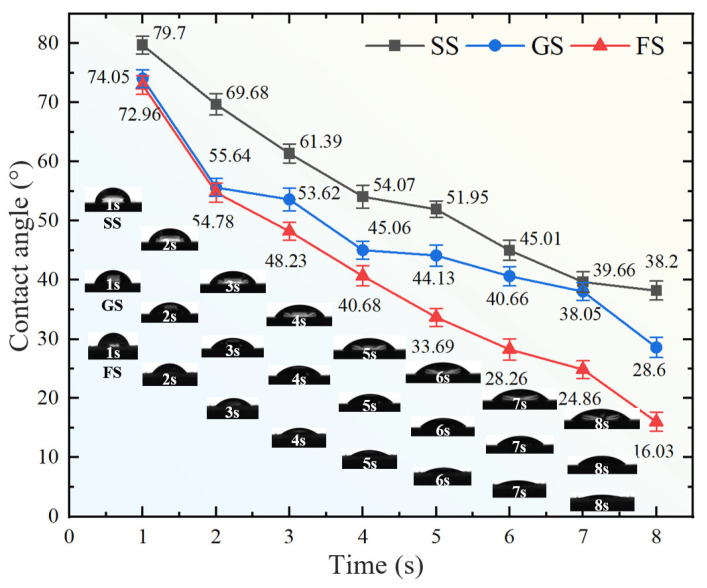
Surface wettability contact angle.

**Figure 7 materials-16-05074-f007:**
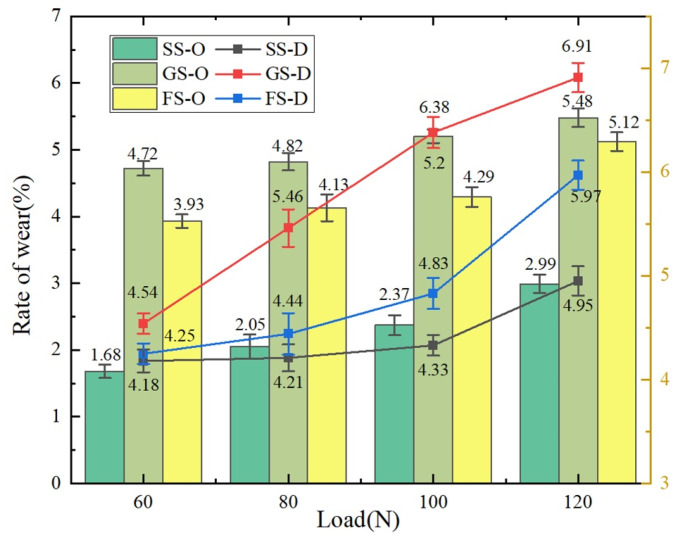
The wear rate of the sample.

**Figure 8 materials-16-05074-f008:**
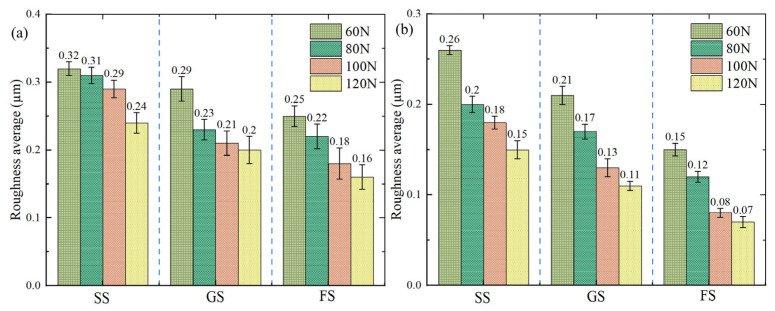
Roughness of sample surfaces under dry friction (**a**) and oil lubrication (**b**).

**Figure 9 materials-16-05074-f009:**
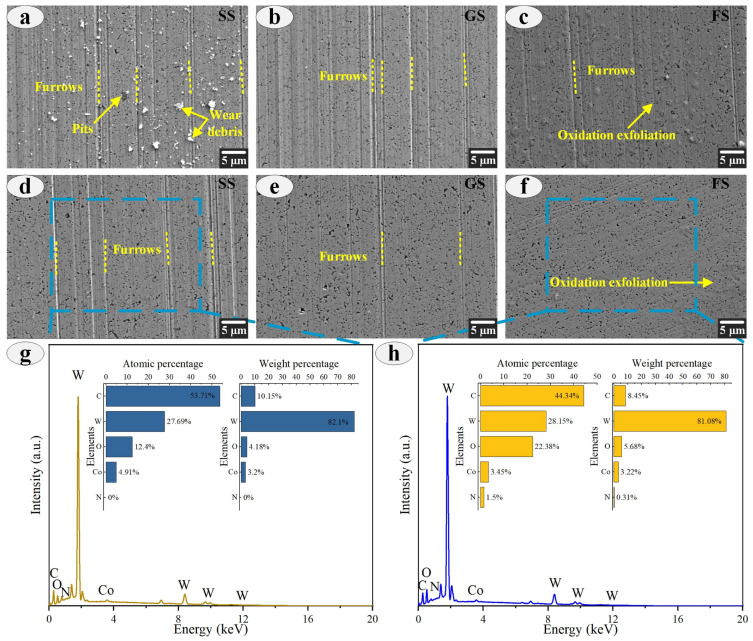
Morphology of sample surface with 120 N load under dry friction (**a**–**c**) and oil lubrication (**d**–**f**); content analyses of SS and FS under oil lubrication (**g**,**h**).

**Figure 10 materials-16-05074-f010:**
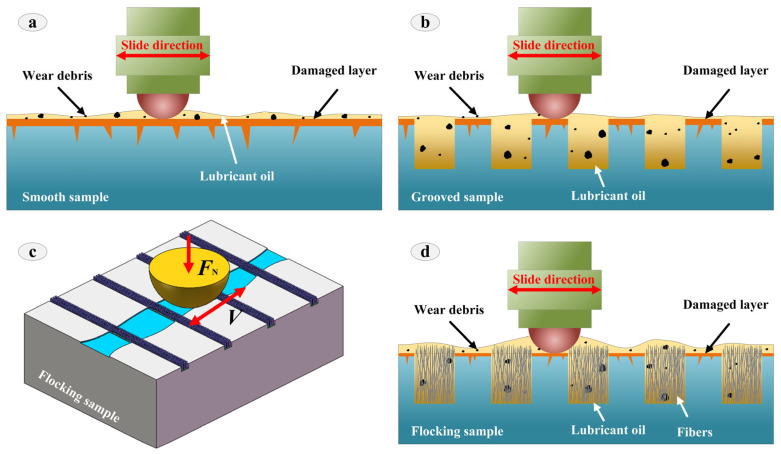
Friction and wear mechanism of SS (**a**), GS (**b**), FS (**d**) during friction process (**c**).

**Table 1 materials-16-05074-t001:** Material properties of nylon.

Density (g/cm^3^)	Melting Point (°C)	Molding Shrinkage (%)	Tensile Strength (Pa)
1.14	260	0.8–1.5	800

**Table 2 materials-16-05074-t002:** Composition and mechanical properties of YG8.

Composition (wt.%)		Density (g/cm^3^)	Impact Energy (J/cm^2^)	Hardness (HRA)	Bending Strength (MPa)
WC	Co				
92	8	14.6–14.9	2.5	89	1500

**Table 3 materials-16-05074-t003:** The processing parameters of the laser.

Laser Wavelength (nm)	Pulse Frequency (kHz)	Average Power (W)	Pulse Width (fs)	Number of Scans
800	5–10	4	100	200

**Table 4 materials-16-05074-t004:** Geometrical parameters of test samples.

Samples	Surfaces	Width (w/µm)	Pitch (d/µm)	Depth (h/µm)
SS	Smooth surfaces	…	…	…
GS	Textured surfaces (Grooves)	200	1000	300
FS	Flocking surfaces (Grooves + Nylon)	200	1000	300

**Table 5 materials-16-05074-t005:** The parameters of electrostatic flocking.

Voltage (V)	Time (s)	Distance (mm)	Fiber Length (mm)	Fiber Diameter (nm)
120	600	30	0.4	67 ± 18

**Table 6 materials-16-05074-t006:** Experimental arrangement.

Samples	SS				GS				FS			
**Lubrication**	No											
**Load (N)**	60	80	100	120	60	80	100	120	60	80	100	120
**Lubrication**	Castor oil											
**Load (N)**	60	80	100	120	60	80	100	120	60	80	100	120

**Table 7 materials-16-05074-t007:** Test conditions for tribological experiments.

Reciprocating Displacement (mm)	Test Time (s)	Test Temperature (°C)	Ambient Humidity (%)
4.15	1200	23 ± 2	50 ± 5

## Data Availability

Data is unavailable due to privacy.

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
