# Peer review of "Study of the Friction Behavior of Embedded Fibers in YG8 Surface Grooves"

_materials, 2023, doi:10.3390/ma16145074_

Round 1
Reviewer 1 Report
The authors have studied the tribological characteristics of YG8 with textured surfaces implanted nylon. Results and discussions are clearly explained. The findings may be useful in engineering applications, which provides theoretical and experimental support for YG8 materials to reduce frictional wear and improve wetting lubrication. In my opinion the manuscript can be accepted after attending the following minor comments to improve the quality of to the presentation.
11. Abstract has to reframed to be in chronological order
22. On Page 6, Para84-86, the sentence has to be rewritten. The paragraph Theoretical studies and mechanistic analyses are conducted for the anti-friction characteristics, wetting characteristics and wear resistance of FS.
33. Figure 4(C), sharp decreasing of friction coefficient of samples (GS-60, 80, 100) was observed in the beginning when compared with sample GS-120, what could be the reason.
44. Few more outcomes can be added in the conclusion part
no comments
Reviewer 2 Report
In the study titled "Tribological characteristics of YG8 with textured surfaces implanted nylon", a lot of effort was spent and the discussion was tried to be comprehensive. However, it is very challenging to understand the main intention of this study.
The title does not reflect the main purpose of the study. as follows; Which materials are used for the purpose of the study? For what purpose is it done? Is the motivation of the work the surface text? its topography? You should make it clear with what purpose and for what purpose your work was done, and use clear, concrete statements that are comforting to the reader. There are some deficiencies in the working texts. For example, there are inconsistencies such as the abbreviations "groove-textured surface (GS)" in the abstract and "surface grooves (GS)" inside. In Figure 6, there is no information about the purpose of the wettability study and how it is done.
Situations like this make it difficult for the reader to understand the subject. The work may need to be carefully prepared and re-evaluated. It seems very difficult to accept in this state.
Reviewer 3 Report
I recommend acceptance "as is".
The main question addressed by the research is how to improve the tribological characteristics (friction and wear properties) of YG8 cemented carbide using textured surfaces and implanted nylon. The study investigates the effects of groove-textured surfaces (GS) and flocking surfaces (FS) prepared on a smooth surface (SS) in combination with nylon (PA) and oil on the friction characteristics of the samples under different applied loads. The research aims to find a solution that reduces friction, minimizes wear, enhances oil storage capacity, and improves the surface profile characteristics of high-hardness materials.
The research topic is relevant in the field of tribology, which is the study of friction, wear, and lubrication of interacting surfaces. By investigating the effect of groove-textured surfaces and implanted nylon, the research aims to address the challenge of surface wear and unstable processes associated with tungsten alloy steel.
The conclusions are consistent with the evidence and arguments presented.
The friction coefficient is found to be positively related to the load due to changes in the contact area between friction pairs. GS causes surface disruptions and tooth jump, while FS exhibits the lowest friction coefficient and load. The combined effect of lubricant and composite structure reduces fluctuations in friction force and load, with FS showing the best anti-friction properties at specific conditions. The wear mechanisms differ among SS, GS, and FS: SS experiences abrasive wear, GS experiences adhesive wear, and FS experiences chemical wear. FS improves friction performance, repairs surface damage, and enhances wear resistance. During sliding, fiber material diffuses and migrates at the friction interface, leading to accelerated oxide layer formation. FS improves surface morphology and demonstrates excellent tribological performance.
The references are appropriate and numerous.
Round 2
Reviewer 2 Report
Confusing situations pointed out in your work have been substantially resolved. When reviewed for the second time, the main purpose of the study emerged and became noteworthy. It is only recommended to consider the minor correction below.
The "In order to find out the influence of oil lubrication on the friction performance of FS, the wettability test of different samples was carried out by contact angle measuring instrument (JC2000D1, Shanghai Zhongchen Digital Technology Equipment Co. LTD, China). The acquisition frequency of the contact intersection measuring device is set to 1s to capture 9 images. The lubricant of FS was completely wetted after 8s, so the images of the first 8s were collected and analyzed." explanation in section “3.1 Wettability behavior of lubricants” is written in “2.2. Experimental procedure” section.
Author Response
Dear editor
Thank you very much for your support and valuable suggestions for our manuscripts. In response to the comments given by the review experts, we changed the title of the article to “Study on the friction behavior of embedded fibers in YG8 surface grooves”. The content of the abstract has been comprehensively modified. The conclusion part of the article has also made some modifications and additions. In addition, we also responded to the other comments given by the three reviewers. At the same time, the content of the text we read in detail after also made some adjustments to the text. These adjustments will not affect the framework and innovation of the text. In order to facilitate the review of editor and reviewers, the red font in the text is the content of the first revision, and the blue font is the content of the second revision.
Thank you very much for your attention and consideration.
Sincerely yours,
Haohan Zhang
